# iPSC-Derived Hereditary Breast Cancer Model Reveals the *BRCA1*-Deleted Tumor Niche as a New Culprit in Disease Progression

**DOI:** 10.3390/ijms22031227

**Published:** 2021-01-27

**Authors:** Lucie Portier, Christophe Desterke, Diana Chaker, Noufissa Oudrhiri, Afag Asgarova, Fatima Dkhissi, Ali G. Turhan, Annelise Bennaceur-Griscelli, Frank Griscelli

**Affiliations:** 1Institut National de la Santé et de la Recherche Médicale–UMR935/UA9, University Paris-Saclay, 94800 Villejuif, France; lucieportier.lp@gmail.com (L.P.); christophe.desterke@inserm.fr (C.D.); diana.chaker@inserm.fr (D.C.); noufissa.oudrhiri@aphp.fr (N.O.); Afag.asgarova@inserm.fr (A.A.); ali.turhan@aphp.fr (A.G.T.); annelise.bennaceur@inserm.fr (A.B.-G.); 2INGESTEM, CITHERA, National IPSC Infrastructure, INSERM University Paris-Saclay, 94800 Villejuif, France; 3Division of Hematology, APHP-Paris Sud University Hospitals, 94270 Le Kremlin Bicêtre, France; 4Institut National de la Santé et de la Recherche Médicale, UMR1082, University of Poitiers, 86000 Poitiers, France; fatima.dkhissi@univ-poitiers.fr; 5Faculty of Medecine, University Paris-Saclay, 94270 Le Kremlin Bicêtre, France; 6Département de Biologie Médicale et Pathologie Médicales, Service de microbiologie, Gustave Roussy Cancer Campus, 94805 Villejuif, France; 7Faculté de la Sorbonne Paris Cité, Faculté des Sciences Pharmaceutiques et Biologiques, University of Paris, 75006 Paris, France

**Keywords:** mesenchymal stem cells, iPSC, BRCA1, breast cancer, angiogenesis, periostin

## Abstract

Tumor progression begins when cancer cells recruit tumor-associated stromal cells to produce a vascular niche, ultimately resulting in uncontrolled growth, invasion, and metastasis. It is poorly understood, though, how this process might be affected by deletions or mutations in the breast cancer type 1 susceptibility (BRCA1) gene in patients with a lifetime risk of developing breast and/or ovarian cancer. To model the BRCA1-deleted stroma, we first generated induced pluripotent stem cells (iPSCs) from patients carrying a germline deletion of exon 17 of the BRCA1 gene (BRCA1+/− who, based on their family histories, were at a high risk for cancer. Using peripheral blood mononuclear cells (PBMCs) of these two affected family members and two normal (BRCA1+/+) individuals, we established a number of iPSC clones via non-integrating Sendai virus-based delivery of the four OCT4, SOX2, KLF4, and c-MYC factors. Induced mesenchymal stem cells (iMSCs) were generated and used as normal and pathological stromal cells. In transcriptome analyses, BRCA1+/− iMSCs exhibited a unique pro-angiogenic signature: compared to non-mutated iMSCs, they expressed high levels of HIF-1α, angiogenic factors belonging to the VEGF, PDGF, and ANGPT subfamilies showing high angiogenic potential. This was confirmed in vitro through the increased capacity to generate tube-like structures compared to BRCA1+/+ iMSCs and in vivo by a matrigel plug angiogenesis assay where the BRCA1+/− iMSCs promoted the development of an extended and organized vessel network. We also reported a highly increased migration capacity of BRCA1+/− iMSCs through an in vitro wound healing assay that correlated with the upregulation of the periostin (POSTN). Finally, we assessed the ability of both iMSCs to facilitate the engraftment of murine breast cancer cells using a xenogenic 4T1 transplant model. The co-injection of BRCA1+/− iMSCs and 4T1 breast cancer cells into mouse mammary fat pads gave rise to highly aggressive tumor growth (2-fold increase in tumor volume compared to 4T1 alone, *p* = 0.01283) and a higher prevalence of spontaneous metastatic spread to the lungs. Here, we report for the first time a major effect of BRCA1 haploinsufficiency on tumor-associated stroma in the context of BRCA1-associated cancers. The unique iMSC model used here was generated using patient-specific iPSCs, which opens new therapeutic avenues for the prevention and personalized treatment of BRCA1-associated hereditary breast cancer.

## 1. Introduction

A tumor is a complex structure composed of malignant cells that are strictly coupled with tumor stroma; the stroma, in turn, contains a wide variety of cells that have been implicated in cancer initiation and development, vascular network growth, local invasion, and metastasis. In breast cancer, this non-malignant compartment can account for up to 70% of the tumor mass, and also plays a pivotal role in resistance to treatment [1].

Recent research has identified mesenchymal stem/stromal cells (MSCs) as important stromal facilitators of tumor development, as well as integral components of the cancer stroma in both experimental and clinical settings [2,3,4,5]. In the context of breast carcinomas, MSCs have been shown to display a pro-malignant phenotype [6,7] and to promote breast cancer progression, invasion, and metastasis [8,9,10,11,12]. Several studies have described a cross-talk between MSCs and breast cancer cells (BCCs): BCCs stimulate secretion of the chemokine CCL5 from MSCs, which then interacts, via paracrine signaling, with the BCC chemokine receptor CCR5 to promote BCC motility, invasion, and metastasis [9]. MSCs have also been shown to cause aberrant expression of micro-RNAs by BCCs, providing BCCs with enhanced cancer stem cell properties [8], and to promote the production of lysyl oxidase by BCCs, which was found to enhance the metastasis of weakly metastatic cancer cells to the lungs and bones [11].

A small fraction of breast cancers (~1–5%) can be attributed to familial mutations in the *BRCA1* and *2* cancer susceptibility genes; such mutations confer an elevated lifetime risk of breast, ovarian, and other cancers. However, the mechanisms underlying tumor formation in these patients are still unknown. BRCA1 is known to be involved in several signaling pathways and cellular processes, such as DNA damage response, regulation of cell cycle progression, apoptosis, and ubiquitination. Pathogenic variants of BRCA1 are created by small insertions or deletions that give rise to a non-functional protein; both alleles of *BRCA1* must be mutated and lost for cancer to develop. Approximately, 85–90% of hereditary *BRCA1*-mutation carriers will develop breast cancer [13] and the majority of these cases demonstrate an aggressive breast tumor phenotype—basal-like, triple-negative {ER(−), PR(−) HER2(−)} [14,15], which is associated with a strong stromal reaction. The mechanism(s) by which *BRCA1*-mutated stromal cells affect the pathogenesis of breast cancer remain to be clarified. It has been shown that phenotypic aberrations arise early in pre-tumoral normal cells with a “first hit” germline mutation. Indeed, even in apparently normal epithelial cells, *BRCA1* heterozygosity induces significant molecular alterations, such as changes in gene expression profiles, the activation status of intracellular signaling cascades (i.e., hormone receptors), degree of differentiation, and genomic stability [16,17]. To date, the molecular and functional consequences of *BRCA1* heterozygosity in stromal cells have been poorly investigated. We hypothesized that mesenchymal stem cells carrying mutations in the *BRCA* cancer susceptibility genes could enable malignant transformation and tumor progression in *BRCA1* carriers. By characterizing the key factors that are altered in *BRCA1*-haploinsufficient stromal cells, it might be possible to identify therapeutic strategies to prevent or treat *BRCA*-associated cancers. To model human stromal cells with a heterozygous *BRCA1* mutation, we used *BRCA1*-mutant human induced pluripotent stem cells (iPSCs) derived from the blood cells of patients with an inherited form of breast cancer. Using these, we are able to report for the first time a strong genotype-to-phenotype correlation, and a relationship between the genetic alteration in MSCs and their influence on the establishment of a permissive vascular niche that promotes tumor growth, invasion, and metastasis. Furthermore, *BRCA1*-mutant iPSC-derived Induced MSCs (iMSCs) models offer a potential framework to evaluate the ability of drugs to modulate the effects of *BRCA1* defects during developmental windows of susceptibility.

## 2. Results

### 2.1. Generation and Characterization of iPSCs and Derivation of Induced Mesenchymal Stem Cells (iMSCs)

To identify early molecular drivers in the establishment of *BRCA1*-deleted/mutated tumors, we generated iPSC lines from the peripheral blood mononuclear cells (PBMCs) of patients who carried a germline deletion of exon 17 of the *BRCA1* gene and who, based on their family histories, were at a high risk for cancer. Specifically, we identified two patients (P1 and P2) who represented two generations of a family with the same *BRCA1* susceptibility gene. Using PBMCs of these two affected family members and two normal (*BRCA1* wildtype) individuals, we established a number of iPSC clones via non-integrating Sendai virus-based delivery of the four Yamanaka reprogramming factors: OCT4, SOX2, KLF4, and c-MYC [18]. In previous publications, we characterized the iPSCs generated from P2 [19] and confirmed the deletion of exon 17 in *BRCA1* by multiplex ligation-dependent probe amplification analysis [20]. All iPSC clones generated from Patient 1 demonstrated human embryonic stem cell morphology and expressed pluripotency factors (NANOG, SOX2 and OCT4) as well as stem cell-specific surface markers (TRA-1-60 and SSEA4) (Appendix A). Importantly, the iPSC lines from both patients had normal karyotypes (Appendix A) and demonstrated the capacity to differentiate into all three germ layers in vivo in teratomas (Appendix A) [19].

Together, these data indicated that somatic cells from BRCA1 patients could be properly reprogrammed, maintain a pluripotent state, and differentiate effectively.

Next, iMSCs were derived from two different lines of *BRCA1* WildType (WT) iPSCs and from the P1- and P2-derived lines of iPSCs (carrying a germline deletion of exon 17 in *BRCA1*; Figure 1A). Both *BRCA1* WT and deleted iPSC-derived iMSCs had typical MSC morphology (Figure 1A); they expressed N-Cadherin and Vimentin mesenchymal markers (revealed via immunostaining; Figure 1B) as well as *Snail1*, *Snail2, N-Cadherin,* and *Vimentin* mRNA (revealed via RT-PCR; Figure 1C). As we had observed for the iPSCs, the karyotypes of all iMSCs were normal (46XX) (Figure 1D). The phenotypes of WT and *BRCA1*-deleted iMSCs were similar: both expressed CD73, CD90, and CD105, and lacked expression of CD34 and CD45 (Figure 1E). An immunoblotting assay revealed that *BRCA1*-deleted iMSCs expressed 50% less BRCA1 protein than WT iMSCs did (ratio of 0.57) (Figure 1F).

### 2.2. Increased Angiogenesis Potential in BRCA1+/− iMSCs

To investigate gene expression differences between iMSC lines, we performed a transcriptome analysis on *BRCA1*+/− (derived from P2 iPSCs) and WT iMSCs. The Significance Analysis for Microarray algorithm was used to identify differentially expressed genes (DEGs) between the two conditions. Of the 261 DEGs highlighted by this analysis, the majority (219) were found to be upregulated in *BRCA1*+/− cells compared to WT. This differential expression profile enabled significant discrimination among *BRCA1*+/− and WT samples by both unsupervised classification (Figure 2A) and principal component analysis (*p*-value = 0.00014, Figure 2B). We next examined the functional enrichment of genes that were overexpressed in *BRCA1*+/− samples using the Gene Ontology (biological process) database; this analysis revealed an important association between these genes and the functions of adhesion, and more particularly, anatomical structure development (Figure 2C). Using these results, we constructed a functional network of all DEGs that were implicated in development (Appendix A), which underlined the importance of developmental genes in this differential gene expression profile. Together, these data suggested that *BRCA1* may play an important role in mesenchymal-derived iPSCs.

Next, we performed a gene-set enrichment analysis on the transcriptome data using the “hallmarks” database; these results highlighted an increase in angiogenesis functionality in *BRCA1*-deleted iMSCs (Normalized Enrichment Score (NES) = +1.64, *p*-value < 0.0001, Figure 3A). A similar analysis performed using the gene set collection from the Gene Ontology database confirmed the relationship between this gene expression profile and angiogenic processes, specifically the functions “sprouting angiogenesis” (NES = +1.91, *p*-value < 0.0001, Figure 3A), “angiogenesis” (NES = +1.51, *p*-value < 0.0001), and “lymphoangiogenesis” (NES = +1.28, *p*-value < 0.0001, Figure 3A). An unsupervised classification based on the differentially expressed angiogenic-related genes enabled the discrimination of *BRCA1+/*− samples from their WT homologs (Figure 3B). Angiopoietin 1 (*ANGPT1*), which is implicated in angiogenesis (Figure 3C), was also considerably upregulated (FC = +28) in *BRCA1+/−* samples compared to WT. Overall, the upregulated genes in the angiogenesis network (Figure 3C) represented diverse classes of molecules, including receptors (*PDGFRB*, *PDGFRA*, *LRP5*, *TEK*), ligands (*ANGPT1*, *PDGFA*, *VEGFB*, *BMP4*, *GREM1*, *THBS1*, *ANG*, *CTGF*), transcription factors (*FOXC2*, *MEIS1*, *MSX1*, *EPAS1*, *HOXB3*, *KLF5*, *ELK3*), and extracellular matrix components (*POSTN*, *LUM*, *MMP2*, *SPP1*) (Appendix A).

### 2.3. Pro-Angiogenic Activity of BRCA1+/− iMSCs Leads to Upregulation of HIF-1α and Angiogenic Factors

HIF-1α is expressed at very high levels under hypoxic conditions and, following translocation into the nucleus, heterodimerizes with HIF-1β to form HIF-1, which initiates the transcription of several angiogenic factors [21]. To investigate the relationship between *BRCA1* deletion and HIF-1α expression, we cultured iMSCs under both hypoxic and normoxic conditions. Regardless of the culture conditions, Western blot analysis revealed that the expression level of HIF-1α was increased in *BRCA1*+/− iMSCs from both patients (P1 and P2). Compared to controls (WT iMSCs and physiological iMSCs isolated from the bone marrow of a healthy donor), HIF-1α by P2-derived *BRCA1*+/− iMSCs increased 2-fold after 8 h of culture in normoxic conditions (Figure 4A) and 2.5-fold after 4 and 8 h in hypoxic culture conditions (Figure 4B). Similarly, for P1-derived *BRCA1*+/− iMSCs, the concentration of HIF-1α increased after 4 and 8 h in hypoxic culture conditions by 5.6- and 3-fold, respectively, compared to WT iMSCs (Appendix A). These results suggested that the loss of one allele of *BRCA1* was able to trigger the stabilization of HIF-1α and/or prevent its degradation in the presence of oxygen. We next wanted to confirm whether the upregulation of HIF-1α in *BRCA1*+/− iMSCs resulted in a corresponding increase in the expression of genes that encode hypoxia-inducible proteins linked to angiogenesis. To verify the angiogenic potential of both WT and *BRCA1*-deleted iMSCs, we quantified the mRNA of angiogenic molecules from the hypoxia-inducible *VEGF*, *PDFG*, and *ANGPT* subfamily and compared them with levels found in primary human umbilical vein endothelial cells (HUVECs). Compared to controls (WT iMSCs and HUVECs), many of these factors were significantly overexpressed in *BRCA1*+/− iMSCs under hypoxic conditions, including *PDGFa*, *VEGFa*, *ANGPT1*, and *ANGPT2,* which, respectively, act as ligands for *PDGFRa*, *KDR/flk1*, *Flt1*, and *TEK/TIE2* (which were also upregulated in *BRCA1*+/− iMSCs under both normoxic and hypoxic conditions; Figure 4C). Interestingly, the expression of *VEGFc*, which mediates lymphangiogenesis, was 2.5- and 2.0-fold higher in *BRCA1*+/− iMSCs under normoxic and hypoxic conditions, respectively, compared to levels in WT iMSCs (Appendix A).

To further evaluate the potential pro-angiogenic activity of *BRCA1*+/− iMSCs, we first performed an in vitro Matrigel angiogenesis assay which evaluated the formation of tube-like structures from HUVECs. With the inclusion of conditioned medium (CM) from *BRCA1*+/− iMSCs that had been cultured in either normoxia (Figure 4D) or hypoxia (data not shown), we observed a significant increase in the growth of tube-like structures on extracellular matrix compared to cultures that used WT-derived CM. The CM from *BRCA1*+/− iMSCs dramatically enhanced the length and number of total branches, with a significant increase in the number of extremities and segments (Figure 4E).

We then used an in vivo method to assess the role of *BRCA1*+/− iMSCs in triggering the development of an efficient network of vessels. To better represent the processes that occur in the human body, we measured angiogenesis in three dimensions using a Matrigel plug angiogenesis assay in mice. Progenitor endothelial cells (PECs) were mixed with equal amounts of *BRCA1*+/+ or *BRCA1*+/− iMSCs and subcutaneously injected in nude mice; after 10 days, the plugs were analyzed. At a macroscopic level, the plugs generated with *BRCA1*+/− iMSCs displayed hemorrhagic areas, whereas their control group counterparts appeared much less vascularized (Figure 4F). Histological analysis of the *BRCA1*+/− iMSC plugs revealed the presence of complete tube-like vessel structures, with lumen that contained a large number of red blood cells (Figure 4F). These kinds of structures were never found in plugs generated with PECs+WT iMSCs or PECs alone. Only the presence of *BRCA1*+/− iMSCs exerted an angiogenic paracrine effect on PECs by inducing the development of large and organized networks of vessels.

### 2.4. BRCA1 Haplo-Insufficiency Induces Upregulation of Periostin and Increases iMSC Migration

In our gene expression analysis, we found that periostin (*POSTN*) mRNA was highly upregulated in *BRCA1*+/− iMSCs (fold change ×8) compared to WT iMSCs. To confirm this result, we compared the amount of *POSTN* mRNA by RT-PCR, and protein levels by Western blot and ELISA using cell supernatants from *BRCA1*+/− and WT iMSCs. *POSTN* mRNA levels were significantly enhanced, with a 2.5-fold change in *BRCA1*+/− iMSCs compared to WT. In contrast, iPSCs produced very low levels of *POSTN* (Figure 5A). After 8 h of culture, soluble POSTN was detected in the supernatant of *BRCA1*+/− iMSCs at levels that were 17-fold higher than those of WT iMSCs (6599 ± 286 vs. 385 ± 50 pg/mL; Figure 5B). Likewise, POSTN expression as measured by Western blot analysis (at 93 kDa) was found to be 2.4-fold higher in *BRCA1*+/− iMSCs compared to WT after 8 h of culture (Figure 5C). Since POSTN is known to activate signaling pathways that promote integrin-dependent cell adhesion and motility, we performed a wound-healing assay with both sets of iMSCs. We made a linear scratch/wound on monolayers of both cell types and measured the gap closure rate at four different timepoints. As expected, *BRCA1*+/− iMSCs migrated faster than WT iMSCs (Figure 5D); in the former group, the wound closed after 24 h, while in the latter a gap was still present at 48 h (Figure 5E). Since the proliferation rate could interfere with measurements of migration, we also performed an MTT (bromure de 3-(4,5-dimethylthiazol-2-yl)-2,5-diphenyl tetrazolium) proliferation assay lasting 72 h. As shown in Figure 5F, the proliferative rate of both iMSCs was identical in culture at a low concentration of serum (Figure 5F). To confirm the functional relevance and involvement of POSTN in the observed cell motility of *BRCA1*+/− iMSCs, we investigated the consequences of POSTN knockdown. We transiently transfected siRNAs directed against POSTN into *BRCA1*+/− and *BRCA1*+/+ iMSCs, and again screened cell motility in a wound-healing assay. We first optimized transfection by using a fluorophore-labeled siRNA (Accell Green Non-targeting Control siRNA); with this, we reached an efficacy of 95.3% after transfection of 2.8 µM of siRNA (Figure 6A). *POSTN* mRNA was inhibited by 90 and 99%, respectively, in *BRCA1*+/+ and BRCA1+/− iMSCs (Figure 6B). After 48 h of culture, POSTN knockdown significantly reduced the cell motility of *BRCA1*+/− iMSCs, which demonstrated an inhibition of 56% compared to POSTN-expressing controls (Figure 6C, Kruskal–Wallis test, *p*-value = 0.000000002162). Instead, POSTN inhibition did not have any effect on the migration of *BRCA1*+/+ iMSCs compared to control cells, revealing that, unlike *BRCA1*-deleted iMSCs, WT iMSCs are not dependent on POSTN for migration (Figure 6C,D). These findings confirm that POSTN produced and secreted by BRCA1-deleted iMSCs has the ability to promote cell motility and potentially the migration and invasion of breast cancer cells.

### 2.5. BRCA1+/− iMSCs Promote Breast Tumor Cell Growth and Metastasis

To investigate the effect of *BRCA1*+/− iMSCs on tumor growth and invasion in vivo, we assessed the activity of 4T1 breast tumor cells in the presence or absence of both types of iMSCs. We tracked metastasis in the lungs by using 4T1 cells that carried a luciferase-expressing cassette (Luc-4T1) which could then be quantified using bioluminescence imaging analysis. Luc-4T1 cells, either alone or pre-mixed with *BRCA1*+/− or *BRCA1*+/+ iMSCs, were injected into the mammary fat pads of NOD-SCID mice; measurements of tumor volume were taken over the next 18 days. A progressive enhancement in tumor growth was detected with both iMSCs, but with a higher rate for *BRCA1*+/− iMSCs compared to WT. On day 18, tumors from the 4T1+ *BRCA1*+/− iMSC co-transplantation group were 2-fold larger compared to the tumors derived from injection of 4T1 cells alone; instead, co-transplantation of 4T1 cells with WT iMSCs did not result in tumors that were significantly larger than those derived from 4T1 cells alone (Figure 7A). Tumor growth was logarithm 10 transformed in order to fit the Gaussian distribution assessed by the Shapiro test (*p*-value = 0.3352, significance superior 0.05 in Shapiro test) and uniform distribution of quantile on QQ-plot (graph not shown). On Gaussian transformed growth data between day 10 and 18, two-way ANOVA analysis was applied, showing a significant effect of the growth (*p* = 0.01283) without interaction of the time effect (interaction *p* = 0.87696). These results implicate *BRCA1*+/− iMSCs in a drastic enhancement of tumor growth, which was confirmed by bioluminescence imaging analysis (Figure 7B). This increase in tumor size was also correlated with the expression of POSTN, as detected by immunohistochemical analyses of tumor sections at day 18 (Figure 7C). Intratumoral angiogenesis, as assessed by CD34 immunostaining of tumor sections, was found to be significantly increased in the *BRCA1*+/− iMSC treatment group (Figure 7C). A well-developed vessel network, with new, completely structured vessels, was detected in the sections of tumors that had been co-injected with *BRCA1*+/− iMSCs (score of 7.3 ± 0.1), with a lower score for 4T1 + WT iMSCs or 4T1 alone (scores of 5.5 ± 0.11 and 2.4 ± 0.11, respectively; Figure 7D, Kruskal–Wallis test, *p*-value = 0.000001292). To test whether *BRCA1*+/− iMSCs, which were enriched in POSTN and angiogenic factors, promoted migration and metastasis, we measured the occurrence of lung metastases in the three groups of mice. Treatment with *BRCA1*+/− iMSCs significantly enhanced the spontaneous metastasis of 4T1 tumors in the lungs 18 days after transplantation, with more numerous and more positive lung areas (6764 ± 275) than in mice treated with WT iMSCs (5243 ± 382) or without any iMSCs (3147 ± 195) (Figure 7E Kruskal–Wallis test, *p*-value = 0.0004595, and F); these results were also confirmed by bioluminescence imaging analysis (Figure 7B).

## 3. Discussion

The aim of our study was to evaluate the effect of *BRCA1*-deleted (*BRCA1*+/−) iMSCs on breast tumor development and progression, compared to iMSCs that contained two copies of *BRCA1* (*BRCA1*+/+ iMSCs). The *BRCA1*+/− iMSCs used here were produced from iPSCs that were generated by reprogramming blood cells (PBMCs) from two patients with a heterozygous deletion of exon 17 of the *BRCA1* gene [19].

Using gene expression analysis, we first identified a set of interrelated gene transcripts whose increased expression in *BRCA1*+/− iMSCs results in the deregulation of several important biological processes, including those implicated in blood vessel development, cell adhesion, and integrin-mediated signaling pathways. Overall, we identified 261 different genes with differential expression between *BRCA1*+/− and *BRCA1*+/+ iMSCs, and these differences enabled significant (*p* = 0.00014) discrimination between the two cell lines. More than 20% of these genes were linked with angiogenesis and sprouting angiogenesis, including *VEGFb*, *PDGFa*, and *ANGPT1*, and their related receptors (*PDGFRB*, *PDGFRA*, *TEK*). Since all of these factors are known to be regulated by HIF-1α [22], we investigated the relationship between *BRCA1* deletion and HIF-1α after having exposed cells to hypoxic or normoxic conditions. Under both culture conditions, Western blots revealed a clear increase in HIF-1α levels in *BRCA1*+/− iMSCs derived from both patients compared to WT iMSCs. These results are in accordance with a previous report that the modification of *BRCA1* in fibroblasts using a shRNA lentiviral approach increased levels of HIF-1α compared to unmodified fibroblasts [23]. In that work, the authors hypothesized that the shBRCA1 treatment decreased levels of succinate dehydrogenase subunit B (SDHB, a complex II subunit) with respect to those of the parental fibroblasts, resulting in an excess of succinate in the cytoplasm that then inhibited prolyl hydroxylase domain (PHD) oxidation of HIF-1α. This in turn led to the stabilization of HIF-1α and to an increase in the half-life of this protein [24]. Similar results have been observed in clinical settings from several cohorts of patients with sporadic and hereditary breast carcinogenesis; specifically, significant overexpression of HIF-1α was reported in *BRCA1*-related cancers [25]. Furthermore, in patients with hereditary breast cancers, positive associations were found between HIF-1α expression and both high tumor grade and shorter relapse-free survival [26]. One potential mechanism to explain these results could be a probable decrease in prolyl hydroxylase enzyme 3 (PHD3), which together with the von Hippel-Lindau tumor suppressor participates in the proteasomal-degradation of HIF-1α [27].

Based on these previous results, we hypothesized that, in response to hypoxia, HIF-1α becomes stabilized in *BRCA1*+/− iMSCs and can translocate into the nucleus in order to regulate a variety of hypoxia-inducible target genes [28]. As expected, in *BRCA1*+/− iMSCs we observed significant upregulation compared to WT of several hypoxia-inducible genes, including *VEFGa*, *VEGFc*, *PDGFa*, *ANGPT1*, and *ANGPT2*. From these results, we concluded that *BRCA1*+/− iMSCs may have more pro-angiogenic potential, which was subsequently confirmed in vitro and in vivo. In vitro, conditioned medium produced from *BRCA1*+/− iMSCs was shown to be more effective in differentiating HUVECs into tube-like structures compared to CM from WT iMSCs. In vivo, *BRCA1*+/− iMSCs were shown to efficiently induce endothelial progenitor cells to form vessel-like structures in a Matrigel plug angiogenesis assay.

By gene expression analysis, Western blot analysis, and RT-PCR, *BRCA1*+/− iMSCs were shown to overexpress periostin (POSTN) compared to WT controls. In addition, an ELISA performed on cell supernatants revealed that only *BRCA1*+/− iMSCs, and not WT, were able to efficiently produce POSTN. POSTN is a secreted cell adhesion protein normally expressed in mesenchyme-derived cells [29]. The N-terminal region of the protein influences cell function, while the *C*-terminal regulates cell-matrix organization and interactions by binding extracellular matrix proteins such as collagen I/V, fibronectin, heparin, and POSTN itself. POSTN binds to integrins through its Fatty Acid Synthase (FAS) domains and activates the Akt/PKB and the FAK-mediated signaling pathways, which together contribute to increased tumor invasion, migration, and metastasis [30]. As POSTN is involved in cell adhesion, motility, and migration, we conducted a wound-healing assay that revealed that *BRCA1*+/− iMSCs have a superior ability to migrate compared to WT iMSCs. Furthermore, this effect was significantly reversed when *BRCA1*+/− iMSCs were transfected with POSTN-specific siRNA. POSTN was previously shown to be involved in the epithelial-mesenchymal transition of carcinoma cells [31,32,33,34,35], a process that is responsible for the dissemination of primary tumor epithelial cells to the sites of metastasis and for the dedifferentiation program that leads to the increased malignant behavior of tumors [36]. In order to determine how both types of iMSCs affect tumor progression, we performed an in vivo study to assess the behavior of breast carcinoma cells when co-injected with iMSCs into the mammary fat pad of NOD-SCID mice. For this, we used a syngenic tumor model based on the triple-negative 4T1 breast tumor cell line, which is known to be highly tumorigenic and invasive, and which, unlike most tumor models, can spontaneously metastasize from the primary tumor in the mammary gland to multiple distant sites including the lung, similar to human mammary cancer [37]. We demonstrated that *BRCA1*+/− iMSCs enhanced 4T1 tumor growth and lung metastasis when co-injected into mouse mammary fat pads; this result was not observed following co-injection of *BRCA1*+/+ iMSCs. These effects were significantly correlated with an increase in intra-tumoral angiogenesis, as assessed by murine CD34 immuno-staining of tumor sections, and with the detection of POSTN immuno-reactive protein by immuno-histochemistry. Together, these results suggest that modulation by *BRCA1*+/− iMSCs, which secrete high levels of POSTN, conferred enhanced self-renewal ability on 4T1 cancer cells, which is one of the characteristics of cancer stem cells. This is consistent with a previous report that one of the effects of POSTN is to increase the prevalence of breast cancer stem cell phenotypes [38]. We also confirmed that the positive effect of *BRCA1*+/− iMSCs on 4T1 tumor growth and dissemination is related to the increased development of tumor vessel networks, which promote lung metastasis. Furthermore, the vessels that developed appeared thicker compared to those obtained from co-injection of WT iMSCs or from the injection of 4T1 cells alone. This provided further evidence that *BRCA1*+/− iMSCs create a more favorable vascularized microenvironment than their *BRCA1*+/+ counterparts. It is probable that this effect is due to factors secreted by *BRCA1*+/− iMSCs that have been shown to influence angiogenesis, including VEFGa, VEGFc, PDGFa, ANGPT1, and ANGPT2. These results reveal for the first time that *BRCA1* haplo-insufficiency in iMSCs strongly affects the tumor microenvironment by promoting tumor angiogenesis, specifically through the release of angiogenic factors under hypoxia and the secretion of POSTN, which promotes angiogenesis through paracrine regulation of the adhesion and migration of endothelial cells. Interestingly, POSTN is also able to regulate the immunosuppressive function of immature myeloid cells in ER-negative breast cancer patients and thus promote premetastatic niche formation in the lungs during breast tumor metastasis [39]. It cannot be ruled out that *BRCA1*+/− iMSCs might also facilitate the progression of lung metastasis by regulating the immunosuppressive function of myeloid cells.

Our findings suggest that POSTN may play a crucial role in the initiation, progression, and dissemination of *BRCA1*-deleted breast cancer, and could serve as a prognostic biomarker in predicting poor survival outcomes in specific subgroups of patients with aggressive and invasive triple-negative breast cancers. Furthermore, our study suggests that POSTN could represent a new potential target for the prevention and treatment of breast tumor metastasis. These results are in concordance with those published by Ryner et al. and Kujawa et al. showing the negative prognostic role of high POSTN level in tumor stroma of ovarian cancer, in high-grade serous ovarian cancer [40,41]. In the future, iPSC-derived *BRCA1*+/− iMSCs could have important applications in high-throughput drug screening and the testing of new drugs to prevent or to cure hereditary breast and/or ovarian cancers by specifically targeting the stromal compartment with germline *BRCA1* mutations.

## 4. Methods

### 4.1. Generation of iMSCs from iPSCs

Induced pluripotent stem cells (iPSCs) were produced and maintained on feeder cells as previously described [18]. Briefly, peripheral blood mononuclear cells (PBMCs) were obtained from two patients who carried a deletion in exon 17 of the *BRCA1* gene; these cells were then reprogrammed into iPSCs using the Oct3/4, Sox2, Klf4, and cMyc transgene factors [18]. These iPSCs exhibit typical markers of pluripotency (Nanog, Sox2, SSEA4, and TRA-1-60) and the ability to differentiate into the three germ layers [19]. To produce Mesenchymal Stem Cells (iMSCs), we used our established protocol, described in [42]. Briefly, iPSCs were detached using DPBS (Life Technologies, Thermo Fisher Scientific, Foster City, CA, USA), supplemented with 0.5 mM EDTA (Life Technologies, Thermo Fisher Scientific, Foster City, CA, USA) and 1.8 mg/L NaCl (Sigma Aldrich, Darmstadt, Germany), and seeded in a Geltrex-coated culture plate with Essential 8 medium (Life Technologies, Thermo Fisher Scientific, Foster City, CA, USA) and 1% penicillin/streptomycin. After 80% confluence was reached, the medium was changed to Minimum Essential Medium alpha (MEM, Gibco, Thermo Fisher Scientific, Foster City, CA, USA) that was supplemented with 10% fetal bovine serum (Hyclone, GE Healthcare, Logan, UT, USA), 1 ng/mL basic fibroblast growth factor, 0.1 mmol/L nonessential amino acids, 1 mmol/L l-glutamine, 0.1 mmol/L β-mercaptoethanol, and 1× penicillin-streptomycin (all from Invitrogen, Thermo Fisher Scientific, Foster City, CA, USA). The medium was changed every day. After 3–4 weeks of culture, iMSCs were characterized by flow cytometry, Western blot analysis, and RT-PCR. To assess the reaction of cultured cells to hypoxic conditions, cells were maintained at 37 °C in 2% O_2_. For some experiments, we used adult bone marrow derived MSCs as additional control samples.

### 4.2. Transcriptome and Bioinformatics Analysis

Total RNA was extracted from iPSC-derived iMSCs using the TRIzol protocol (Ambion, Thermo Fisher Scientific, Foster City, CA, USA) according to the manufacturer’s instructions. The integrity of RNA was assessed on a Bioanalyzer 2100 (Agilent Technologies, Santa Clara, CA, USA). RNA material that passed quality control was used to create aRNA probes via T7 RNA amplification following the manufacturer’s instructions (Affymetrix, Thermo Fisher Scientific, Foster City, CA, USA). aRNA-labeled microarray probes were hybridized on an Affymetrix human gene microarray, version ST2.0. Following extraction of the signal on an Affymetrix scanner, the matrix was normalized using the RMA algorithm with Expression Console version 1.4.1.46 (Affymetrix, Thermo Fisher Scientific, Foster City, CA, USA). Transcriptome data were submitted to the Gene Expression Omnibus database under accession number GSE104693. The Significance Analysis for Microarray algorithm [43] was used to identify differentially expressed genes between iMSCs derived from BRCA1+/− iPSCs and iMSCs derived from WT iPSCs; genes with a fold change higher than 2 (in absolute value) were retained after applying a False Discovery Rate (FDR) threshold of less than 5 percent. An unsupervised principal component analysis was performed with the FactoMineR R-package [44]. Functional enrichment was analyzed with the standalone software Goelite version 1.5 [45]. Gene-set enrichment analysis (GSEA) was used to identify differences between the two iMSC populations [46] using the Reactome and Gene Ontology biological process gene sets from MSigDB version 1.5. We specifically examined angiogenesis- and lymphangiogenesis-related gene sets that were enriched in BRCA1+/− iMSC samples compared to WT iMSCs (positive Normalized Enrichment Score with *p*-value *p* < 0.05 and FDR < 0.25). The enriched genes in each angiogenic gene set were pooled in order to perform unsupervised classification (complete distances–Euclidean metric) with the package gplots in R version 3.0.2. Angiogenic genes that were upregulated in *BRCA1*+/− iMSC samples and identified as enriched by the GSEA analysis were also used to build a functional enrichment network with Cytoscape software version 3.0.2: octagon elements represent the enriched gene set, circle elements represent corresponding enriched genes, edges represent links between genes and functions (gene sets). The size of the octagon indicates the number of connected genes and the color scale (from yellow to purple) represents the Normalized Enrichment Score (NES) obtained in the GSEA analysis [47].

### 4.3. Matrigel Angiogenesis Assay

One-hundred microliters of Matrigel was added to a 48-well culture plate and incubated at 37 °C for 1 h. Human umbilical vein endothelial cells (HUVECs) were seeded at a density of 6 × 10^4^ cells per well in 280 µL of EBM-2 on Matrigel; to this we added 70 µL of conditioned medium (CM) recovered from cultured *BRCA1*+/+ or *BRCA1*+/− iMSCs in normoxic or hypoxic conditions. Prior to addition, CM was concentrated through Amicon tubes (Millipore, Darmstadt, Germany) by centrifugation at 3400× *g* for 20 min. After 18 h of incubation under normal culture conditions, vascular tube formation was examined by phase-contrast microscopy and scored using the Angiogenesis Analyzer plugin of ImageJ software (NIH, Bethesda, MD, USA).

### 4.4. RT-qPCR Analysis

Total RNA was extracted using the TRIzol protocol (Ambion, Thermo Fisher Scientific, Foster City, CA, USA) according to the manufacturer’s instructions and treated with DNAse 1 (Invitrogen, Thermo Fisher Scientific, Foster City, CA, USA). RNA was reverse-transcribed to obtain cDNA using an AffinityScript Multiple Temperature cDNA Synthesis Kit (Agilent, Santa Clara, CA, USA) according to the manufacturer’s instructions. Real-time qPCR reactions were performed using FastStart Universal SYBR Green Master Mix (Roche, Mannheim, Germany) on an Agilent Mx3005p apparatus (Santa Clara, CA, USA). The thermal cycling conditions comprised an initial denaturation step at 95 °C for 10 min and 50 cycles of 95 °C for 15 s and 65 °C for 1 min. Results were visualized using MxPro software (Agilent, Santa Clara, CA, USA) and analyzed with the 2ΔΔCt method.

### 4.5. Immunoblot Analysis

Cells were lysed on ice with 1x RIPA buffer (Pierce, Thermo Fisher Scientific, Foster City, CA, USA) that was supplemented with protease inhibitors (Complete™ Mini EDTA-free Protease Inhibitor cocktail tablets, Roche, Mannheim, Germany), then centrifuged at 12,000× *g* for 20 min at 4 °C. Cell lysates were separated by electrophoresis on 4–20% Tris-Glycine gel (Invitrogen, Thermo Fisher Scientific, Foster City, CA, USA) and transferred onto a nitrocellulose membrane. After blocking with TBS Tween 5% BSA for 1 h at room temperature with gentle shaking, membranes were incubated with primary antibodies at 4 °C overnight. The primary antibodies used were rabbit monoclonal anti- HIF-1α (1:1000, #14179, Cell Signaling, Danvers, MA, USA, clone D2U3T), rabbit polyclonal anti- POSTN (1:10,000, ab83739, Abcam, Cambridge, UK), and mouse monoclonal anti- β-actin (1:50,000, A3854, Sigma, Aldrich, Darmstadt, Germany, clone AC-15). They were detected using HRP conjugated secondary antibodies and the signal was revealed by chemiluminescence with SuperSignal West Dura or Femto reagents (Thermo Fisher Scientific, Foster City, CA, USA). Images were acquired with a G:BOX iChemi Chemiluminescence Image Capture system (Syngene, Synoptics Ltd, Cambridge, UK) and analyzed with ImageJ software. For BRCA1 detection, cells (4 × 10^6^) were lysed with 100 µl NuPage™ LDS Sample Buffer (Thermo Fisher Scientific, Foster City, CA, USA). Equivalent amounts of protein of each sample were analyzed by SDS Page; samples were transferred onto nitrocellulose membranes using the iBlot Dry Blotting system (Thermo Fisher Scientific, Foster City, CA, USA) and probed with mouse anti-BRCA1 monoclonal antibody (1:50, OP92, Merck Millipore, Darmstadt, Germany, clone MS110) or anti-β-actin antibody (Sigma-Aldrich, Darmstadt, Germany). Immunodetection was performed using the chemiluminescent substrate Luminata Forte (Merck Millipore, Darmstadt, Germany). The intensity of bands was quantified by computer-based densitometry of Western blot bands (ImageJ software).

### 4.6. ELISA

Supernatants of cultured iMSCs were harvested after 1 h, 4 h, and 8 h of culture in both normoxic and hypoxic culture conditions. To determine the amount of human periostin produced in the supernatant of iMSCs, we used a sandwich-based ELISA assay according to the manufacturer’s instructions (DuoSet ELISA DY3548B, R&D Systems, Biotechne, Minneapolis, MN, USA). Absorbance was detected at 450 nm using a Modulus II microplate multimode reader (Turner Biosystems, Sunnyvale, CA, USA).

### 4.7. Immunofluorescence

Cells were fixed in 4% PFA for 15 min at room temperature. After washing with PBS, cells were permeabilized when necessary with PBS that contained 0.25% Triton X-100 for 10 min. The primary antibodies used were rabbit monoclonal anti-*N*-Cadherin (1:200, #11039-H08H, Sino Biological, Beijing, China, clone 020) and mouse monoclonal anti-Vimentin (1:200, # 562338, BD Pharmingen, Franklin Lakes, NJ, USA, clone RV202) and were incubated in PBS with 1% BSA for 45 min at room temperature. After washing three times with PBS that contained 0.1% Triton, cells were incubated for 45 min at room temperature with fluorochrome-conjugated anti-rabbit and anti-mouse secondary antibodies diluted in PBS with 1% BSA. Finally, nuclei were stained with DAPI (1 μg/mL) and the coverslips were mounted with Dako Mounting Medium (Dako, Agilent Technologies, Santa Clara, CA, USA).

### 4.8. Flow Cytometry

The phenotype of iMSCs was characterized and analyzed by flow cytometry. Briefly, iMSCs were suspended in PBS and incubated for 45 min at 4 °C in the dark with the following antibodies: mouse monoclonal anti-CD34-APC (1:10, #555824, BD Biosciences, San Jose, CA, USA, clone 581), mouse monoclonal anti-CD45-PerCP (1:10, #130-094-975, Miltenyi Biotec, Bergisch Gladbach, Germany, clone 5B1), mouse monoclonal anti-CD73-PE (1:10, #550257, BD Biosciences, San Jose, CA, USA, clone AD2), mouse monoclonal anti-CD90-PE (1:10, PN IM1840U, Beckman Coulter, Indianapolis, IN, USA, clone F15-42-1-5), and mouse monoclonal anti-CD105-PE (1:10, PN A07414, Beckman Coulter, Indianapolis, IN, USA, clone 1G2). Stained iMSCs were then washed with PBS and analyzed by flow cytometry (MACSQuant, Miltenyi Biotec, Bergisch Gladbach, Germany).

### 4.9. Transfection of iMSCs with siRNA

POSTN expression was knocked down in iMSCs using siRNA. iMSCs were cultured in six-well plates until 80–90% confluence was reached, then transfected with 2.8 µM of POSTN siRNA (Thermo Fisher Scientific, Foster City, CA, USA) using the Viromer Blue transfection reagent (Lipocalyx, Halle, Germany) according to the manufacturer’s instructions. As a nonspecific control, scrambled sequence siRNA was used at the same concentration. In order to assess transfection efficiency, we used GFP siRNA (Dharmacon, Lafayette, CO, USA) in the same conditions.

### 4.10. Wound Healing Assay

To investigate the migration ability of the iMSCs, they were first plated in six-well plates and allowed to grow into monolayers. A linear wound was made on cell monolayers with a sterile pipette tip. Phase-contrast microscopy images were regularly taken and the wound area was measured by analyzing the images using ImageJ software. In experiments using POSTN-knockdown iMSCs, the transfection protocol with POSTN siRNA was performed twelve hours prior to the wound healing assay.

### 4.11. MTT Assay

The iMSCs were seeded into 24-well plates at a density of 1.5 × 10^4^ cells/well. After 24, 48, and 72 h of culture, cell viability was tested: we removed half of the medium and added 20 μL of 5 mg/mL tetrazolium salt solution (Sigma-Aldrich, Darmstadt, Germany) to each well, then incubated the plates at 37 °C for 4 h. After removing the medium, formazan crystals were dissolved with DMSO and the absorbance was read at 570 nm with a Modulus II microplate multimode reader (Turner Biosystems, Sunnyvale, CA, USA).

### 4.12. In Vivo Matrigel Plug Assay

Human progenitor endothelial cells (PECs) were obtained from cord blood as previously described [48]. Matrigel plugs were prepared on ice with PBS as control or by mixing 1.5 × 10^6^ PECs with 1.5 × 10^6^ iMSCs, then re-suspended in 200 µL phenol red-free Matrigel (BD Bioscience, San Jose, CA, USA). The mixture was implanted in the backs of 8-week-old male athymic nu/nu mice (Janvier Labs, Le Genest-Saint-Isle, France) by subcutaneous injection using a 25-gauge needle. After 10 days, the plugs were removed, fixed, and embedded in paraffin for histological analysis.

### 4.13. Animal Studies

For in vivo experiments, we used the 4T1 cell line as a breast cancer model, which is an aggressive triple-negative murine cancer cell line. The animal study was conducted according to the guidelines of the Declaration of Helsinki, and approved by an Ethics Committee. We created 4T1-Luc-GFP clones by transducing the cells with retroviruses that harbored a Luciferase-GFP cassette. *BRCA1*+/+ or *BRCA1*+/− iMSCs (3 × 10^5^) were mixed with 4T1-Luc-GFP cells (10^5^) and re-suspended in 100 µL of PBS (ratio: 3:1). The mixture was injected into the fat pad of anesthetized NOD-SCID mice (Taconic, Doussard, France), one injection per mouse [49]. In addition, iMSCs and 4T1-Luc-GFP cells were also injected alone as controls. Tumor growth was measured twice per week and IVIS analysis was performed on the day of sacrifice (day 18). Primary tumors and lung metastases were removed for histological and immuno-histochemical analysis.

### 4.14. Immuno-Histochemistry

Samples of paraffin-embedded sections of fat-pad 4T1 tumors and lung 4T1 metastases were stained to determine the expression levels of POSTN and GFP proteins. After deparaffinization and rehydration, antigen retrieval was performed in 10 mM sodium citrate for 15 min using a pressure cooker. Sections were blocked with 3% hydrogen peroxide for 10 min, followed by protein blocking for 1 h with Power Vision IHC/ISH Super Blocking reagent (Leica Biosystems, Wetzlar, Germany). Sections were incubated with the primary antibodies, rabbit polyclonal anti-POSTN (1:100, ab83739, Abcam, Cambridge, UK) and rabbit polyclonal anti-GFP (1:1000, ab290, Abcam) for 1 h at room temperature. For detection of the primary antibodies, poly HRP anti-mouse/rabbit IgG (Leica Biosystems, Wetzlar, Gerlany) was used. All slides were counterstained with Mayer’s hemalum solution. For GFP staining, images were taken and analyzed using ImageJ software to determine the area of lung metastasis. To quantify the number of intra-tumoral vessels in 4T1 tumors, paraffin sections were prepared by murine CD34 immunostaining as previously described [50]. Digital images of immunohistochemically stained tissue slides were obtained at ×20 magnification using a slide scanner (NanoZoomer 2.0-HT, Hamamatsu, Japan). Scanned slides were uploaded onto the Calopix database (Tribvn Healthcare, Chatillon, France).

### 4.15. Statistical Analysis

Each experiment was performed at least twice. Statistical significance was evaluated using 2-sided Student’s t-test and compared with a non-parametric Kruskal–Wallis test for low sample numbers by groups. Additionally, a two-way ANOVA was applied to tumor growth data transformed in logarithm base 10 after assessing the Gaussian distribution through the Shapiro test and visualization of uniform distribution on quantile-quantile plot (QQ-plot). Barplots present mean and standard deviation as error bars and boxplots present interquartile ranges.

## Figures and Tables

**Figure 1 ijms-22-01227-f001:**
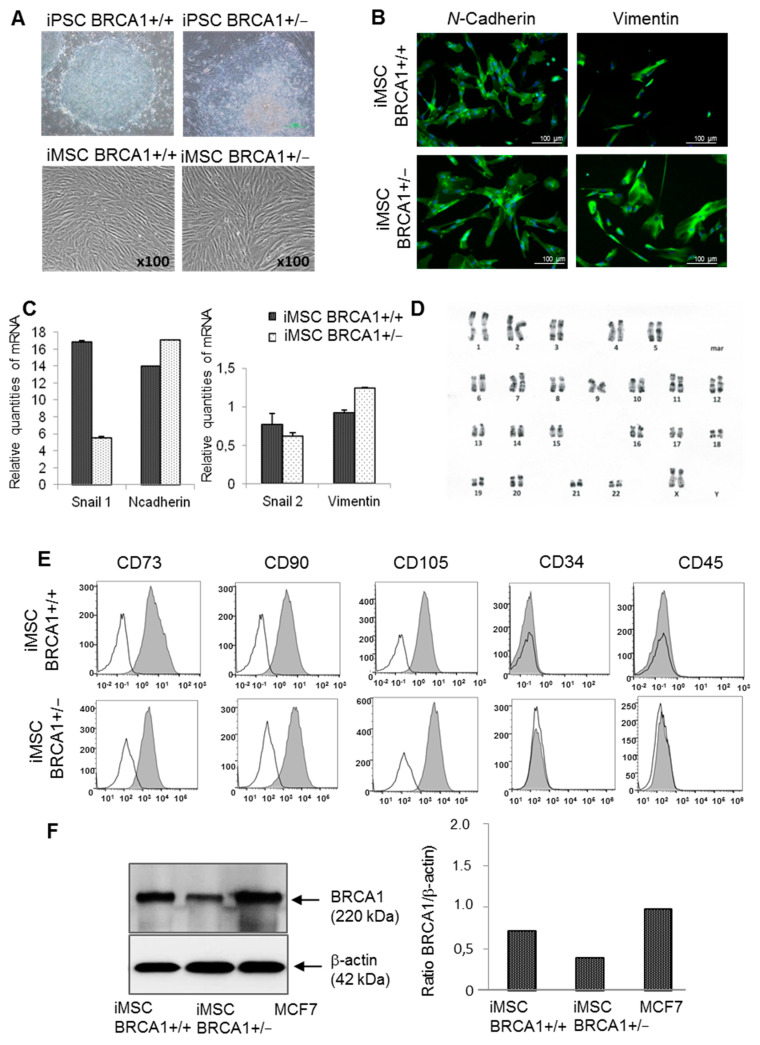
Characteristics of induced pluripotent stem cell (iPSC)-derived *BRCA1*+/− induced mesenchymal stem cells (iMSCs) and *BRCA1*+/+ iMSCs. (**A**) Morphology of iPSCs and iPSC-derived iMSCs, characterized by spindle-shaped cells. (**B**) Immunofluorescence images of iPSC-derived iMSCs stained with *N*-Cadherin and Vimentin mesenchymal markers. (**C**) Expression of mesenchymal markers (Snail 2, *N*-Cadherin, Vimentin, and Snail 1) in both cell lines as determined by RT-qPCR. (**D**) *BRCA1*-deleted iMSCs exhibit a normal karyotype. (**E**) Expression of characteristic MSC markers by flow cytometry. White histograms indicate negative expression. (**F**) Western blot of BRCA1 protein levels in deleted *BRCA1*+/− and wild-type *BRCA1*+/+ iMSCs. Immunoblotting revealed a decrease in BRCA1 protein level in the *BRCA1*-mutated MSCs.

**Figure 2 ijms-22-01227-f002:**
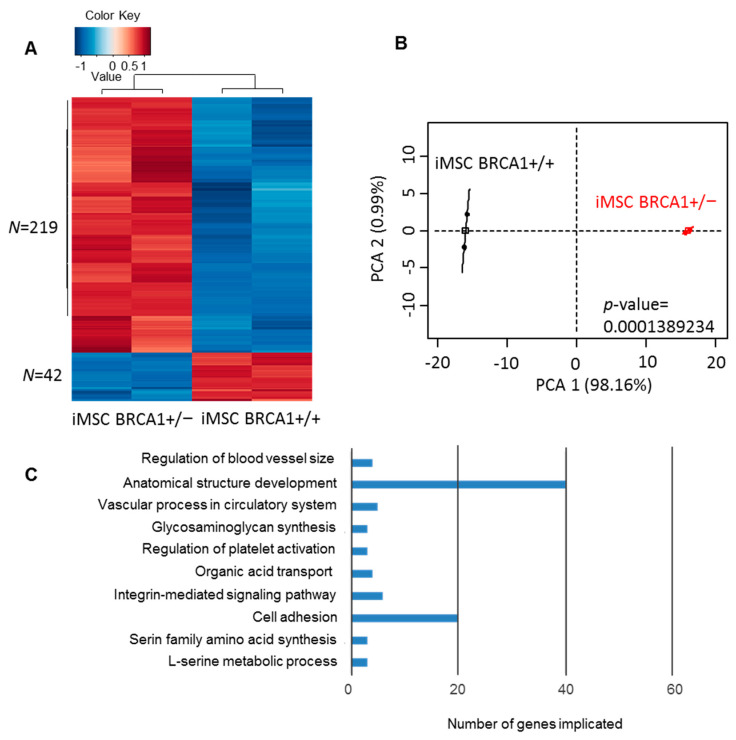
Differentially expressed genes between *BRCA1*+/− and *BRCA1*+/+ iMSCs. (**A**) Transcriptome heatmap with unsupervised classification of genes with differential expression between *BRCA1*+/− and *BRCA1*+/+ iMSCs. (**B**) Principal component analysis of genes with differential expression between *BRCA1*+/− and *BRCA1*+/+ iMSCs. (**C**) Functional enrichment in Gene Ontology Biological Processes of genes that were upregulated in *BRCA1*+/− iMSCs compared to *BRCA1*+/+ iMSCs.

**Figure 3 ijms-22-01227-f003:**
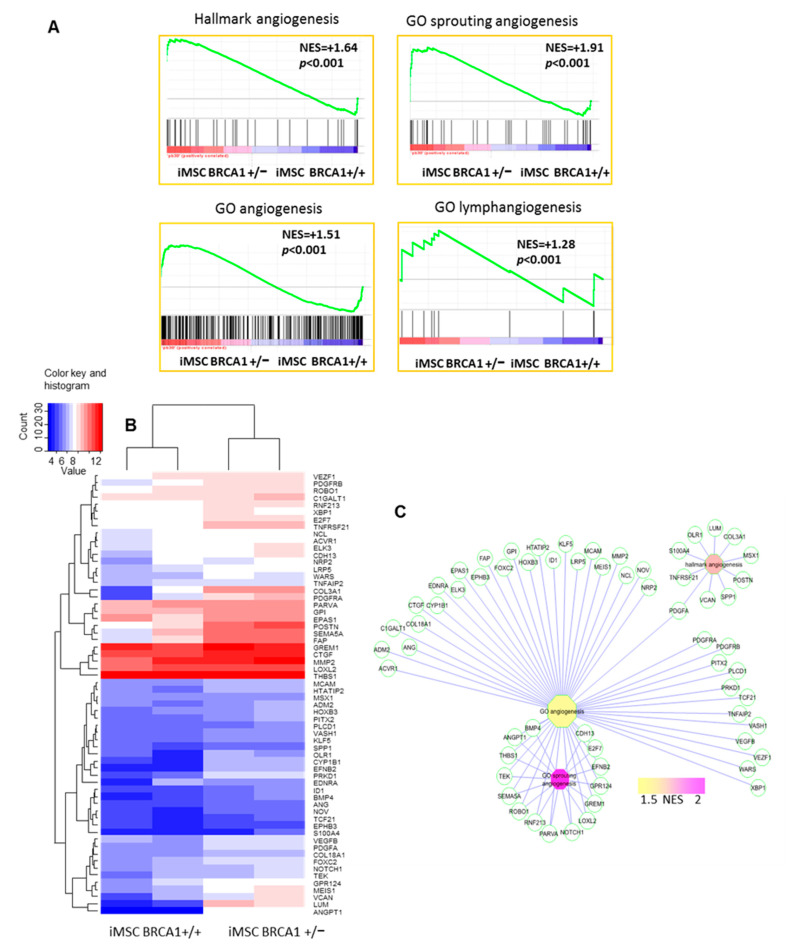
Angiogenesis-associated activity in the transcriptome of *BRCA1*+/− iMSCs. (**A**) Enrichment in angiogenesis- and lymphangiogenesis-related gene sets from the Hallmarks and Gene Ontology Biological Process gene set collections from MSigDB version 5.2 (NES: Normalized Enrichment Score). (**B**) Unsupervised clustering (Euclidean distances) of angiogenic genes that were upregulated in *BRCA1*+/− iMSCs. (**C**) Angiogenesis-related functional network created with genes upregulated in *BRCA1*+/− iMSCs compared to WT (NES: Normalized Enrichment Score).

**Figure 4 ijms-22-01227-f004:**
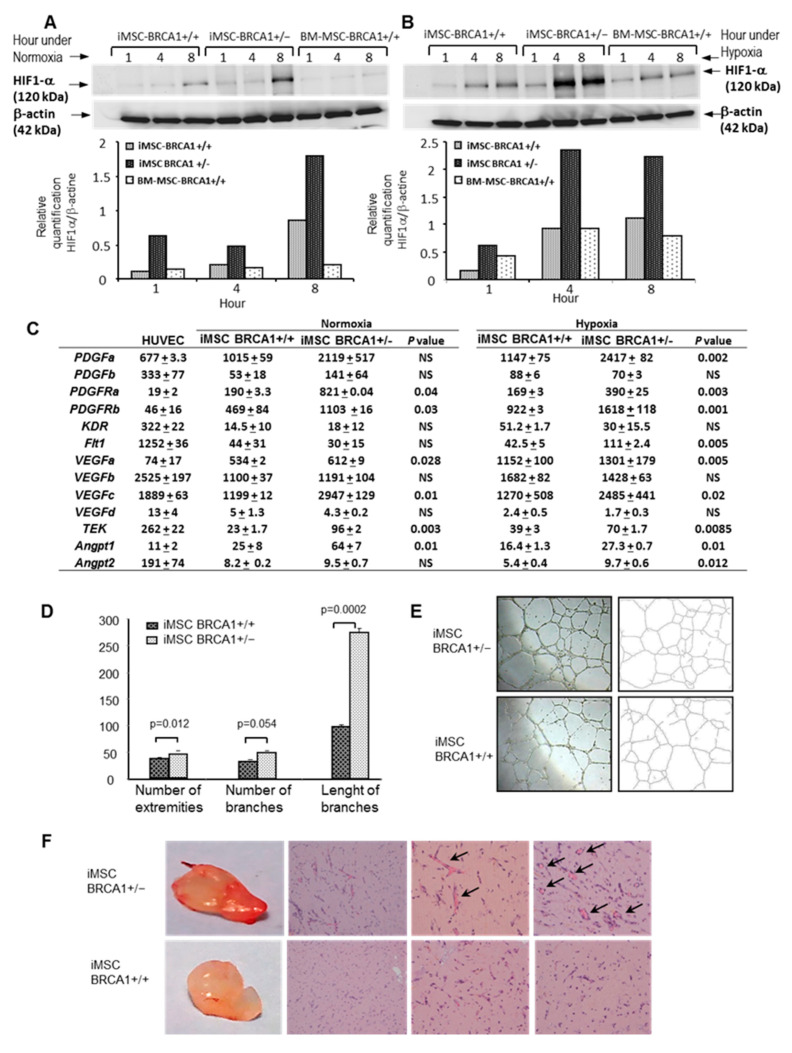
Pro-angiogenic properties of *BRCA1*+/− iMSCs (**A**) Western blot of relative HIF-1α levels in *BRCA1*+/− iMSCs, *BRCA1*+/+ iMSCs, and bone marrow-derived MSCs at 1, 4, and 8 h of culture under normoxia. (**B**) Quantification of HIF-1α levels relative to β-actin in *BRCA1*+/− iMSCs, *BRCA1*+/+ iMSCs, and bone marrow-derived MSCs at 1, 4, and 8 h of culture under hypoxia. (**C**) Expression of a variety of genes encoding hypoxia-inducible proteins in human umbilical vein endothelial cells (HUVECs) and in *BRCA1*+/− and *BRCA1*+/+ iMSCs cultured for 4 days under normoxia and hypoxia, as determined by RT-PCR. (**D**) Matrigel angiogenesis assay using HUVECs and conditioned medium from *BRCA1*+/− or *BRCA1*+/+ iMSCs, showing differences in the number of extremities and the length and number of branches. (**E**) Representative images of tube-like structures on an extracellular matrix. (**F**) Plug angiogenesis assay performed with *BRCA1*+/− and *BRCA1*+/+ iMSCs. Macroscopic image of the plug after 10 days is shown as well as histological analysis after hematoxylin and eosin staining. Plugs created with *BRCA1*+/− iMSCs had complete tube-like vessel structures, with lumen containing a large number of red blood cells (arrows).

**Figure 5 ijms-22-01227-f005:**
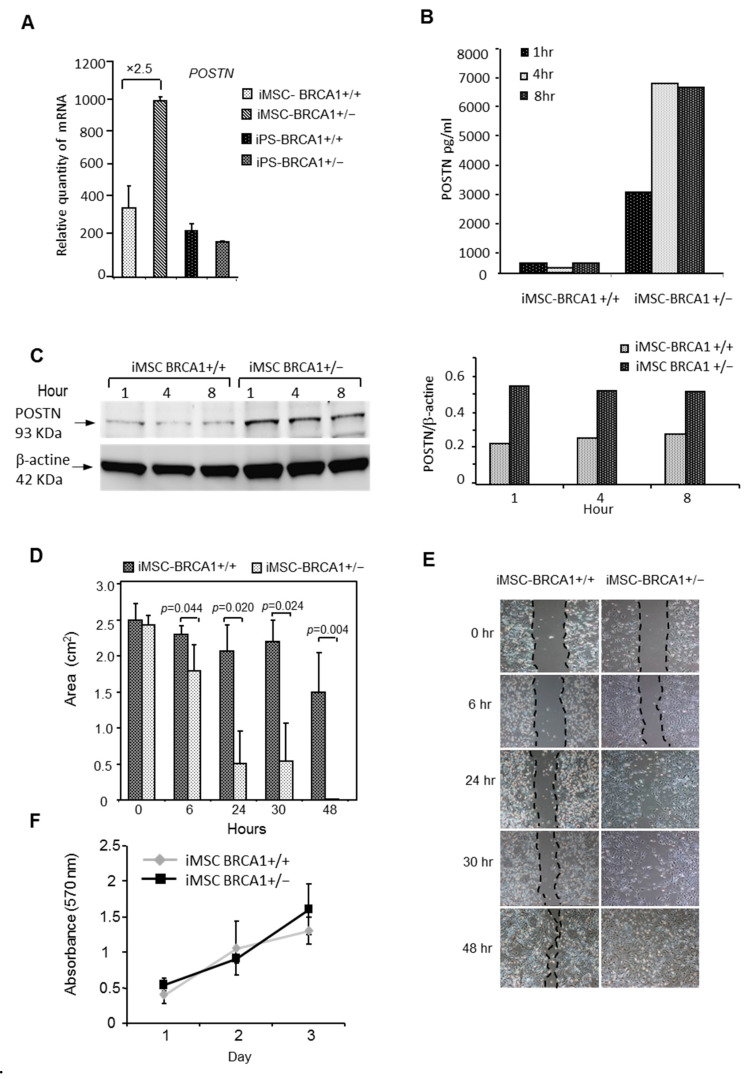
Periostin (POSTN) expression and wound-healing assay. (**A**) Quantitative RT-PCR for *POSTN* expression in iMSCs and iPSCs. (**B**) ELISA for POSTN in the supernatant of *BRCA1*+/− and *BRCA1*+/+ iMSCs after 1, 4, and 8 h of culture. (**C**) Quantification of POSTN levels relative to -actin in *BRCA1*+/− and *BRCA1*+/+ iMSCs after 1, 4, and 8 h of culture under normoxia. (**D**) Cell migration ability measured by a wound-healing assay. Compared with control cells (*BRCA1*+/+ iMSCs), the migration distance of *BRCA1*+/− iMSCs was significantly (2-sided Student’s t-test) greater. (**E**) Representative photomicrographs of the gaps for *BRCA1*+/− and *BRCA1*+/+ iMSCs in culture at different time points (6, 24, 30, 48 h) after scratching in a wound healing assay. (**F**) Proliferation assays using an MTT test on *BRCA1*+/− and *BRCA1*+/+ iMSCs after 1, 2, and 3 days of culture.

**Figure 6 ijms-22-01227-f006:**
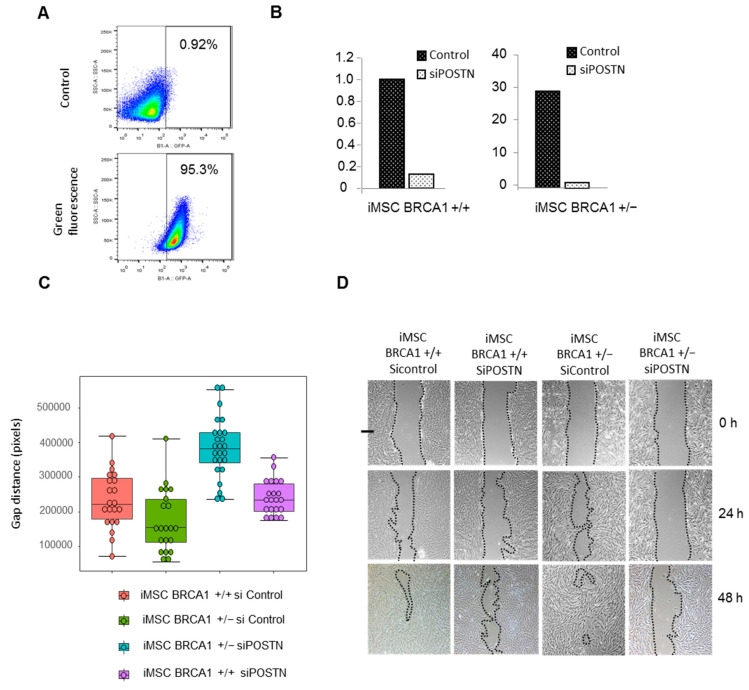
Wound-healing assay with *BRCA1*+/− and *BRCA1*+/+ iMSCs after transfection with siRNA against POSTN. (**A**) Evaluation of the transfection efficacy in *BRCA1*+/− iMSCs using fluorophore-labeled siRNA. (**B**) Knockdown of *POSTN* mRNA in both cell lines as evaluated by quantitative RT-PCR. (**C**) Cell migration ability was measured at 48 h with a wound-healing assay using cell lines transfected with *POSTN* siRNA and siRNA control. (**D**) Representative photomicrographs of the gaps for transfected *BRCA1*+/− and *BRCA1*+/+ iMSCs in culture at different time points (24 and 48 h) after scratching in a wound-healing assay.

**Figure 7 ijms-22-01227-f007:**
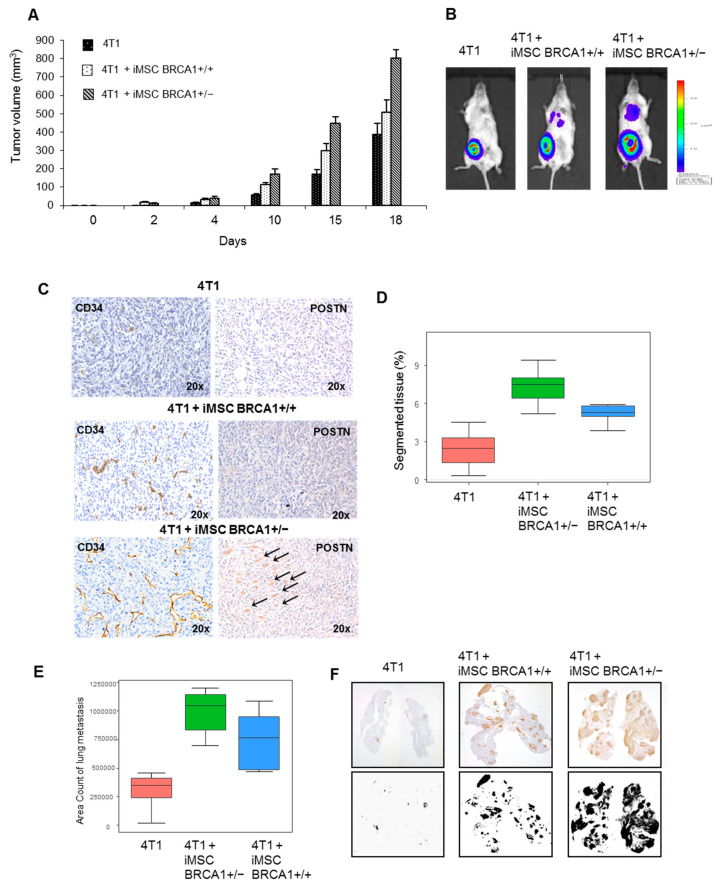
*BRCA1*+/− iMSCs enhance 4T1 tumor formation and lung metastasis. (**A**) To evaluate the in vivo tumorigenic potential of 4T1 cells, a total of 10^5^ Luc-expressing 4T1 cells, either alone or premixed with 3 × 10^5^
*BRCA1*+/− or *BRCA1*+/+ iMSCs, were injected into the mammary fat pad. Tumor growth was recorded. Data are presented as the mean+SD. (**B**) Representative images of tumorigenesis visualized by bioluminescence imaging at day 18. (**C**) Left: CD34 immunostaining of tumor sections, showing a well-developed vessel network with a complete structure of new vessels within *BRCA1*+/− iMSC co-transplanted 4T1 tumors as compared with those co-injected with WT iMSC or only 4T1, right: immunohistochemistry of POSTN-immunoreactive material in tumor sections, showing cells positive for POSTN (arrows). (**D**) The extent of intratumoral vascularization was assessed by CD34 immunostaining and quantified by assessing the percentage of segmented tissue area that was positive for CD34 at day 18. (**E**) Quantification of lung metastatic areas at day 18. (**F**) The extent of lung metastasis is shown after 18 days within the whole lung area following immunohistochemical staining with GFP.

## Data Availability

Transcriptome experiments performed during this study were deposited in open access at NCBI Gene Expression Omnibus database under the access number GSE104693, weblink: https://www.ncbi.nlm.nih.gov/geo/query/acc.cgi?acc=GSE104693 (accessed on 26 January 2021).

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
