# Peer review of "iPSC-Derived Hereditary Breast Cancer Model Reveals the BRCA1-Deleted Tumor Niche as a New Culprit in Disease Progression"

_ijms, 2021, doi:10.3390/ijms22031227_

Round 1
Reviewer 1 Report
The authors have accurately replied to all suggestions and the paper has improved considerably
Author Response
English language and style have been checked and corrected
Reviewer 2 Report
Big thanks to Authors for their effort to adhere to the Reviewers comments. In my opinion the manuscript has been significantly improved.
I have only three remarks to improve:
Figure 1 (E): unit numbers which describe the curves are invisible.
Figure 1 (F): descriptions under the graph (ratio BRCA1/β-actin) should be as follows: iMSC BRCA1 +/+, iMSC BRCA+/-, MCF7, it will be consistent with WB results on the left.
Please pay attention to the same typestyle and its size (there are differences in an abstract, an introduction and in lines 248 to 250), and to descriptions under the figures.
Author Response
Improvements were made as requested
Point 1: Figure 1 (E): unit numbers which describe the curves are invisible.
Figure 1 (E) is modified as requested
Point 2: Figure 1 (F): descriptions under the graph (ratio BRCA1/β-actin) should be as follows: iMSC BRCA1 +/+, iMSC BRCA+/-, MCF7, it will be consistent with WB results on the left.
Figure 1 (F) is modified as requested
Point 3: Please pay attention to the same typestyle and its size (there are differences in an abstract, an introduction and in lines 248 to 250), and to descriptions under the figures.
The typestyle and its size have been corrected in line 248 to 250. the manuscript has been checked concerning typestyle and its size.
Reviewer 3 Report
The authors have satisfactorily addressed the concerns of the reviewers and substantially improved the manuscript.
Author Response
English language and style have been checked and corrected
This manuscript is a resubmission of an earlier submission. The following is a list of the peer review reports and author responses from that submission.
Round 1
Reviewer 1 Report
In this paper Portier et al., investigated on important topic regarding disease progression in hereditary breast/ovarian cancer. The personalisation of breast cancer care is a major factor in improving outcomes and then therefore the discovery of new biomarkers in the era of precision medicine is a current and extremely important challenge.The study is complex due to the extent of the experiments conducted and is suitable for a reader with a background and knowledge in this area. However, I believe it has been clearly presented.
I suggest to the authors few minor revisions:
Methods Section
While "Transcriptonme and Bioinformatics Analysis" is well descripted, "Statistical Analysis" section (Line 612-614) is poor and incomplete.
- Why do you perform unilateral Student's t test? This in un uncommon choice: Because the one-tailed test provides more power to detect an effect, you may be tempted to use a one-tailed test whenever you have a hypothesis about the direction of an effect. Before doing so, authors should consider the consequences of missing an effect in the other direction and if these consequences are negligible and in no way irresponsible or unethical, then they can proceed with a one-tailed test. Therefore, it is important that authors justify this choice.
- In Results sections several times (for example line 262-263, 326-327...) data were summarized with mean +/- standard deviations. Authors must be specified in Statistical Analysis section how data were presented and why they chose mean+/- SD instead of median (interquartile range). Did authors verified normally distribution of tested data? Please integrate the section with these information
- Authors (line 613-614) asserted that they used only Student's t test: this test is not correct when the comparisons are more than two, as it happens in different situations of analysis in this paper. For example in Figure 7A authors compared 4T1 vs 4T1+ iMS BCRCA1+/- vs 4T1 +iMSC BRCA1+/+: t- student test is not appropriate: authors should perform One-Way Anova test ( if data were normally distributed or Kruskall Wallis test for non parametric data ) and a post-hoc analyis with p-value adjusted for multiple comparisons. All analysis in which there were three groups should be corrected using adequate statistical tests (in addition to Figure 7A, also Figure 7D, Figure 7E, Figure 6D) and then authors should insert in statistical analysis the tests performed
Figures
In some figure authors reported p-values, in other only the asterisk symbol. Please make the figures homogeneous: it is more appropriate to report the real value of the p-value rather than the asterisk, or otherwise authors should be report the legend for * that now it is not present.
Reviewer 2 Report
Overall, the work is very valuable and shows that microenvironment of the tumor is an indispensable element of tumor progression. The authors did a lot of interesting experiments. This work requires some corrections to improve comfort for reader and to achieve good quality of presentation of scientific results. The images in supplement are clearly described and are of better quality in comparison to those included main manuscript. So please, provide better quality images and graphs and amend their descriptions to make them more clear. Please check and correct symbols used when you mention HIF1 α (lines 194, 196, 198, 200, 202, 366 etc.) and β actin (line 223) (remove additional signages). Please pay attention to correct gene symbols. Please decide to use one abbreviation of protein mass (you used kD, Kda and KDa). I propose kDa.
Specific remarks:
There is a mess in the Abstract. Authors achieved a lot of interesting results which should be well described in abstract as well. Moreover, they mention about transwell migration test, which wasn’t used in this work.
Figure 1 (A): to make it more clear please add following descriptions: „iMSC BRCA+/-„ under the image of BRCAdel17iPSC and „iMSC BRCA+/+” under the image of wild type iPSC
Figure 1 (F): descriptions under the graph (ratio BRCA1/β-actin) are mixed, they should be as follows: iMSC BRCA1 +/+, iMSC BRCA+/-, MCF7
Figure 1 (D) and (E) should be of better quality (karyotype is unclear and axes depiction of histograms are not visible at all).
Figure 2 (D) should be of better quality and it could be moved to suplement (to consider). Such figures are decorative but provide low scientific value, in fact
Figure 3 (B) and (C) very unclear gene symbols
Please provide the full list of angiogenic genes (from Fig 3B) with their fold change values (to be included in suplement).
The sentence in lines 201 to 204 suggests that HIF1 was secreted from cells and was detected in medium (the phrase „cell secretions of HIF-1” is misleading).
In the sentence lines 204 to 206 (ending with „Supplement Figure 2” phrase) you should better clarify, because 3-fold increase is for 8 h and 5.6-fold increase is for 4 h.
In the chapter „Pro-angiogenic activity of BRCA1+/- iMSCs leads to upregulation of HIF-1 and angiogenic factors” there are strange characters after „HIF-1” name (possibly resulting from conversion of word file into .pdf). Please pay attention and correct.
There are mistakes in abbreviations of some genes in Table in Fig 4 (C) for example PDFG and PDFGR instead of PDGF and PDGFR. Please check and correct.
Fig 4 (E) and (F) presents very valuable images for performed experments but they are of poor quality.
Fig 6 (C) incorrect units at y-axis (100000, 200000 etc.) .
Fig 7 (C) and (F) please improve the quality of images.
Figure 1 in suplement: incorrect description (there is no E image, there is „A & B” instead of „A” on picture).
Methods:
You used phrase „Matrigel tube formation assay” in the method chapter while „matrigel angiogenesis assay” in main text. Please be consistent.
Please describe dilution of antibodies and catalog numbers for all Abs used in the study. Please use following scheme (example): anti-BRCA1 monoclonal antibody (1:75, OP92, Merck Millipore, clone MS110).
Line 405 – you could mention also the works showing negative prognostic role of high POSTN level in tumor stroma of ovarian cancer, particularily in high-grade serous ovarian cancer (HGSOC) which is frequently associated with BRCA1/BRCA2 dysfunction (mutation, LOH, promoter methylation). Please mention following works:
https://pubmed.ncbi.nlm.nih.gov/31936272/
https://pubmed.ncbi.nlm.nih.gov/25838397/
Reviewer 3 Report
Overall, this article is well written, provides a solid foundation and logic for the majority of experiments that were executed. This work contributes to the growing area of research that combines the tumor microenvironment and its interaction with BRCA1+/- status using an elegant model system of induced mesenchymal stem cells derived from patients. My impression is that the this work is of high interest to the IJMS audience and its impact is appropriate for publication upon minor and major revisions. My recommendations are listed below.
Minor Revisions:
- Please provide higher resolution images for the data in every figure. Many panels appear blurry and the text is difficult to read. For example, the text in Figure 2D and Figure 3B is not legible.
- I recommend that the RT-PCR data to be normalized so that the control (iMSC BRCA+/+) is set at 1 and convert the +/- sample relative to 1. This will enable the consolidation and facilitate the comparability of the Y-axis.
- Please edit in parts of the text where the symbol for alpha in HIF1 is converted to a swirl symbol. Please change to a proper alpha symbol
Major Revision:
In the introduction, line 71-74, the authors explain that MSC's have the capacity to induce "weakly metastatic cancer cells" to metastasize to lungs and bones. In the in vivo experiments presented in Figure 7, the authors chose to employ 4T1 cells with iMSC mixtures to test if the BRCA+/- can enhance their tumorigenic potential. However, 4T1 cells are already a highly aggressive cell line with stem cell properties, and have the capacity to metastasize to the lung even when mice are injected with as few at 104 cells. There are many publications that use 104 4T1 cells and in 4 weeks, there are significant lung metastases. In the present study, the authors used 105 cells mixed with iMSCs 1:3 so it is not surprising that this lung metastasis and increase in tumor volume was obtained faster. This does not mean that 4T1 cells alone do not have properties of stemmness. In short, just because they grew faster does not mean it was because they became more stem-like. Indeed, the authors recognize the highly tumorigenic and metastatic potential of 4T1 cells in the discussion so it is problematic to use this model and to conclude that they obtained "more" self-renewal properties when mixed with +/- iMSCs.
To strengthen their argument, I recommend that the authors test in vivo an additional and alternative cell line that is not inherently stem-like and highly aggressive such as the 67NR cell line. Using these cells may take longer than 18 days, even with 105 cells, but this would provide stronger evidence to demonstrate the influence that the BRCA-/+ stromal cells alone exert on tumor cells and may have the capacity to induce weakly metastatic cells.
